# Extracorporeal Shock Wave Therapy as a Helpful Method for Rapid Osseointegration of Dental Implants: Animal Study

**DOI:** 10.3390/biomimetics8020137

**Published:** 2023-03-27

**Authors:** Amir Jafarpour Mahalleh, Ali Hossein Mesgarzadeh, Seyedhosein Jarolmasjed, Abbas Soltani Somee, Monireh Khordadmehr, Yashar Rezaei, Solmaz Maleki Dizaj, Shahriar Shahi

**Affiliations:** 1Student Research Committee, Tabriz University of Medical Sciences, Tabriz 13147-15311, Iran; 2Department of Oral and Maxillofacial Surgery, Faculty of Dentistry, Tabriz University of Medical Sciences, Tabriz 13147-15311, Iran; 3Department of Clinical Sciences, Faculty of Veterinary Medicine, University of Tabriz, Tabriz 51666-16471, Iran; 4Department of Physiotherapy, Faculty of Rehabilitation Sciences, Tabriz University of Medical Sciences, Tabriz 13147-15311, Iran; 5Department of Pathobiology, Faculty of Veterinary Medicine, University of Tabriz, Tabriz 51666-16471, Iran; 6Department of Dental Biomaterials, Faculty of Dentistry, Tabriz University of Medical Sciences, Tabriz 13147-15311, Iran; 7Dental and Periodontal Research Center, Tabriz University of Medical Sciences, Tabriz 13147-15311, Iran

**Keywords:** dental implants, extracorporeal shockwave therapy, bone regeneration, osseointegration, bone volume fraction

## Abstract

The aim of this study was to assess the multi-phasic use of extracorporeal shock wave therapy (ESWT) as an adjuvant treatment to accelerate the osseointegration of titanium dental implants. Initially, twelve titanium mini-screws were inserted in femur bones of six New Zealand rabbits in three groups; the one-time treated group, the three-time treated group, and the control group (without ESWT). Then, 1800 focused shockwaves with an energy flux density of 0.3 mJ/mm^2^ in every phase were used. Fourteen days after the last phase of ESWT, the animals were sacrificed to assess the osseointegration of screws via micro-computed tomography scan (micro-CT scan), biomechanical pull-out test, and histopathological analysis. Pull-out and histopathology analysis showed that the ESWT significantly increased bone regeneration and osseointegration around the implants compared to the control group (*p* < 0.05). Moreover, the pull-out test confirmed that the three-time treated screws needed more force to pull the bone out compared to the other two groups (*p* < 0.05). The mean bone volume fraction between the control group, the one-time treated group, and the three-time treatment group were not statistically significant (*p* > 0.05) according to the micro-CT scan results. Based on our results, ESWT can be suggested as a non-invasive and cost-effective adjuvant for osseointegration of dental implants. However, more in vivo studies and clinical trials are needed for validation of this finding.

## 1. Introduction

Recently, quick progress is seen in dental implant technology and dental implants have been extensively applied in clinical preparations. Dental implants are believed to be the most effective alternative to replace missing teeth. It has long been affirmed that osseointegration of implants is one of the most important factors in obtaining a satisfactory outcome with titanium implants [1,2]. However, numerous problems, comprising a lack of osseointegration, may happen after dental placement.

A proper function and an appealing consequence for a dental implant are related to good bone quality and suitable bone mass at the site of implantation. However, alveolar bone inadequacy may occur due to periodontal disease, trauma, and resorption. Therefore, some helping processes such as bone augmentation surgery (such as guided bone regeneration (GBR)) are essential to protect the bare surface of the implant [3]. Yet, the osseointegration period in the bone defect area is so long. This process is also strictly related to the supply of peripheral blood and the migration and differentiation of osteoblastic cells in the bone marrow [4]. Indeed, the inflexibly and rapidly creating osseointegration in the bone defect zone around the implant is still a main clinical issue.

Once the implant is located in the bone in such a position that it is “well-seated” is known as primary stability. This permits the implant to mechanically familiarize to the host bone until secondary stability is attained. Reduced primary implant stability has been revealed to endanger the osseointegration procedure. Other factors including the density and dimension of the bone surrounding the implant, the implant design, and surgical technique are main factors for the osseointegration procedure as well [5].

Some systemic diseases such as diabetes mellitus and osteoporosis are from main risk factors that influence the achievement rate of dental implant cures [6]. Such diseases are frequently attended by diverse degrees of bone remodeling disorders, which can delay the osseointegration of implants throughout the healing period [6]. An absence of osseointegration directly origins implant loss and surgical disappointment, and is problematic to forecast. Therefore, discovering of novel approaches to advance the osseointegration of implants basically and rapidly, and shortening the healing cycle have become the focus of current clinical research [7].

Osseointegration is a dynamic host-implant interaction that is followed by a local inflammatory process within the implant-tissue interface, and if it happens ideally, it leads to the ideal integration of the implant; which is commonly defined as the absence of any relative movement between the implant and bone surface [8]. To date, numerous studies have been carried out to facilitate dental implant osseointegration in terms of time and quality. Nevertheless, the studies to find the most efficient way to facilitate osseointegration, still go on [9]. The happening of osteointegration failure after dental establishment is often complex and irregular, and current treatment approaches cannot opposite osteointegration failure to attain the best situation.

An easy operation, noninvasive, cost-effective, fast mechanotherapy is probable to resolve this issue. Extracorporeal shock wave therapy (ESWT) is extensively applied to treat late healing, bone nonunion fractures, femoral head necrosis, and the other orthopedic diseases and shows an important part in endorsing bone regeneration [10,11].

Shockwaves (SWs) are sound waves with extremely high pressure and speed produced by underwater high-pressure explosions and vaporization. They have the characteristics of high peak pressure (up to 100 MPa or higher), a fast pressure rise (<10 Ns), short duration (<10 Ms), and a wide frequency range (16–20 MHz) [12]. ESWT has been widely used for multiple medical purposes such as orthopedics and rehabilitation medicine [13], especially bone healing [14]. Investigators instigated discovering their therapeutic potential. Throughout the 1990s and early 2000s, many investigation teams shared results signifying that ESWT could be used as a therapeutic instrument. Investigators established the use of ESWT to decrease pain and encourage healing in bone, tendon, ligament, and fascia in patients with musculoskeletal disorders, and to decrease spasticity in patients with neurological disorders [15].

In the past decades, ESWT showed a positive effect on mandibular bone formation via promoting immune response and secretion of cytokines such as BMP-2 [16]. Previous studies have also shown that ESWT can improve the osteogenic effect of osteoblasts [17], as well as inducing the formation of new blood vessels to accelerate hard and soft tissue healing [18]. According to reports, ESWT can endorse bone formation and osseointegration of titanium devices in vivo [11]. In earlier studies, ESWT has shown good results in regulating the activity of inflammatory cells, osteoblasts, and mesenchymal stem cells [19]. Reports have also stated the part of ESWT in encouraging angiogenesis and bactericidal action [20]. Therefore, extracorporeal shock wave therapy can ease the understanding of osteointegration by regulating the immune response, encouraging regeneration of the jaw and alveolar bone, and endorsing angiogenesis and bactericidal effectiveness [19,20].

The aforementioned characteristics of ESWT are in favor of suggesting ESWT as an effective and non-invasive adjuvant treatment to accelerate the osseointegration of dental implants. Several experiments have specifically considered the effect of ESWT on osseointegration of titanium screws inserted in living bone and reported successful results, more experiments are needed to find the exact details of ESWT protocol in order to facilitate osseointegration of titanium components inserted in living bone [21].

Focused ESWT includes the usage of acoustic waves transmitted in a narrow or focused form. It was first utilized in the early 1980s and appeared as a noninvasive treatment recognized as lithotripsy to eliminate kidney stones [15]. In the decades that followed, the advent of lithotripsy, investigators initiated testing other potential clinical applications for focused ESWT and another form of shock wave therapy known as radial ESWT. Unlike focused ESWT, radial ESWT acoustic waves are transmitted in a more diffuse, radial form [15].

ESWT medical strategies could develop a novel beneficial plan to immunomodulate osseointegration with the last aim of improving clinical achievement and dropping the number of problems in dental implant treatment [10]. Holfeld et al. proposed that the automated stimuli produced by ESWT originate an increase in the permeability of the cell membrane, triggering the release of cytoplasmic RNA over an active procedure reliant on exosomes. Then, the RNA can stimulate the TLR3 in healthy neighboring cells. TLR3 is a portion of the innate immune system and moderates inflammation by the stimulation of the creation of several cytokines. However, the signal transduction mechanism of TLR3 receptors has not yet been clarified [22].

ESWT expanded increasing attention from investigators and clinicians in growing the zones of potential usage in recent years. It is a non-invasive, safe, low-cost, and rapid application method. A sequence of investigations displayed a positive result of ESWT in different tissues such as cardiovascular, neural, and skin [11,22]. It has been reported that the effect of ESWT is dose-dependent and site-specific [11]. Then, other usage and doses used in different parts of the body cannot be useful for another. Extensive investigational and clinical examinations are vital to control suitable shock wave factors. The dose–effect, and safety–efficacy rules needed for optimal therapeutic results should be performed for it [11].

In this study, the effect of focused extracorporeal shock wave therapy on osseointegration of titanium mini-screws as a representative of dental implants on cortical rabbit bone was tested.

## 2. Materials and Methods

Six adult male New Zealand rabbits were purchased from the Pasteur Institute of Iran (Tehran, Iran) and were housed and supervised by the Faculty of Veterinary Medicine, University of Tabriz, Tabriz, Iran.

### 2.1. Inclusion Criteria

-Physical health and maturity of animals-Minimum body weight of 2.0 kg in each animal

### 2.2. Exclusion Criteria

-Any signs of physical imperfections in animals after surgery or during the shock wave therapy procedure.-Significant decrease of body weight in animals at day 28 post-surgery, compared to day 0.-Any imperfections during the surgery process that interfered with the purpose of the study.

Animals received standard food pellets and ad libitum water and were kept under climate-controlled conditions (21 °C; 12 h light/12 h darkness). At the average age of 18 weeks (average weight of 2.5 kg) and 14 days of acclimatization, each rabbit received two sterile self-tapping mini-screws in each cortical femur bone, leaving 12 independent experiment sites. Experiment sites were evenly divided into three groups; a one-time treated group (which underwent Shock Wave Therapy once), a three-time treated group (which underwent shock wave therapy three times), and the control group. Subsequently, on day 7, the one-time treated group and the three-time treated group underwent shockwave therapy (ESWT). In three-time treated group two additional stages of SWT at days 10 and 13 post-surgery were conducted. The control group remained without ESWT treatment and was left to process the osseointegration spontaneously.

On day 28 of the post-surgery, animals were sacrificed by an overdose of sodium pentobarbital (100 mg/kg IP injection). Afterward bone sites were analyzed using a micro-CT scan, the mechanical pull-out test and the histology images [23].

### 2.3. Surgical Process

Surgeries were conducted by general anesthesia (2% Isoflurane, USP, Terrel TM). Legs were shaved accordingly from ankle to hip, and after a lateral skin incision on each leg, a dissection of soft tissue and underlying fascia was made. The periosteum was not manipulated. Bone drilling was performed unicortically at the center of the femoral diaphysis with a 6 × 1.8 mm bur (Drendel + Zweiling 018). Thereafter, a cortical self-tapping mini-screw 6 × 2 mm (Titanium Alloy, Ti6Al4V grade 5, Imen Ijaz Co., Tehran, Iran) was inserted unicortically.

Mini-screws had a core diameter of 1.5 mm and a thread diameter of 2 mm. As mentioned earlier, drilling was performed with a 1.8 mm diameter bur which gave a 0.3 mm wider hole for a clinically used standard 2 mm mini-screws. This was made to have a slight micro defect between screw threads and bone surface, giving a better opportunity to assess the effect of ESWT on bone formation in all three groups. All screws showed a primary stability right after insertion (Figure 1).

Eventually, the facia and skin were sutured in layers using SupaSil Braided Silk 2/0 USP. Animals were given a single dose of antibiotics (Pen-Strep 40 mg/kg penicillin subcutaneously) and an analgesic (Meloxicam 1 mg/kg subcutaneously) right after surgery, and one dose per day for three days after surgery.

### 2.4. Focused Extracorporeal Shock Wave Therapy

Focused ESWT includes the application of acoustic waves transmitted in a narrow or focused design [15]. Seven days after surgery, the first phase of extracorporeal shock wave therapy (Enraf-Nonius Shockwave) was accomplished in both intervention groups which were the one-time treated group and the three-time treated group.

To treat the screw on the right leg, the animals were placed on their left dorsal-lateral side, and the inverse for the screws were inserted in the left leg. The ultrasonic gel was applied on the shaven area of the leg, then an applicator of 3.5 cm diameter was used to give 1800 focused shock waves with an energy flux density of 0.3 mJ/mm^2^, and an energy level of 10 kV [24] on the area. While treating with ESWT, the applicator was moving slowly around its perimeter to ensure that all of the screw-bone interface receives an equal amount of treatment. The three-time treated group received two more phases of ESWT at days 10 and 13 post-surgery, with the exact details as in phase 1. While screws in the control group were left to process the healing and osseointegration spontaneously and did not receive any treatment.

To prevent any discomfort and pain in animals, each animal that was treated with ESWT was given a single dose of meloxicam (0.6 mg/kg, subcutaneously) right after treatment in every phase.

### 2.5. Micro CT-Scan

Samples were made by sawing the bone with a 1.5 cm distance with the screw, which made an approximately 3 cm length of each sample. An in vivo X-ray Micro-Computed Tomography (micro-CT) scanner (LOTUS inVivo, Behin Negareh Co., Tehran, Iran) was used to obtain the best possible image quality.

The scanning parameters were as follows: scanning voltage 50 kVp, current 120 μA, power 8 W, system resolution selected as medium resolution (scanning rotation 360°, acquisition of 1000 projection images).

The frame exposure time was set to 2 s by 2.7 magnification. The total scan duration was 49 min. Slice thicknesses of reconstructed images were set to 20 μm. All the protocol settings process was controlled by LOTUS-inVivo-ACQ software (Behin Negareh Co., Tehran, Iran). The acquired 3D data was reconstructed using LOTUS in vivo-REC by a standard Feldkamp, Da-vis, Kress (FDK) algorithm. Further, LOTUS inVivo-3D was used for rendering of reconstructed images and by adding bone analysis plugin (BAP) inside the software we reported bone volume (BV), total volume (TV) parameters.

### 2.6. Biomechanical Pull-Out Test

Right after micro-CT scanning, samples were taken to the Biomaterials Department of Tabriz Faculty of Dentistry to undergo a pull-out test and measure the quantitative osseointegration in inserted fixtures in all three groups (Hounsfield Universal testing machine). A stainless-steel arch wire (0.5 mm, Ispringen, Remanium, Germany) was used to fix the screws to the tensile grip. For each test, a wire was made dead under flame temperature then it was twisted one round below the screw head as in Figure 2 at strain rate of 0.5 mm/min to gain stable support of the screw whilst performing the test.

The pull-out test was performed at the displacement rate of 0.5 mm/min, and the pull-out force was determined in Newton (N).

### 2.7. Histopathology Evaluation

Histological measures purpose to show good quality sections that can be used for a light microscopic assessment of human or animal tissue [25]. Usually, tissues are immobile with neutral formalin, fixed in paraffin, and then manually sectioned with a microtome to obtain sections. Dewaxed sections are then stained with hematoxylin and eosin (H&E) or can be used for other purposes (special stains, immunohistochemistry, in situ hybridization, etc.). Throughout this procedure, numerous stages and actions are critical to confirm standard and interpretable sections [25].

In this study, the tissue samples were fixed in 10% formalin for 48 h and decalcified in 10% nitric acid and 14% diamino tetra acetic acid. The specimens were routinely processed, embedded in paraffin, sectioned with a microtome in 5-μm slices, and stained with common Hematoxylin and Eosin (H&E). Then, the tissue sections were studied by a light microscope (Olympus, Tokyo, Japan). The presence or absence of inflammation (periostitis, osteitis, and osteomyelitis), bone necrosis (osteosis), and new bone formation were reported for all three experimental groups.

Samples were labeled via three digits. The first digit represents the type of intervention (“0” as the control group, “1” as one-time treated screws, and “3” as three-time treated screws), while the two other digits represent the animal. For example, sample “330” indicates the three-time treated screw on animal number 30 (left leg “3” and right leg “0”).

During performing all of the tests, technicians were blind to the sample labels that they were testing. At the end of each test, quantitative results were directly given to the analyst in the form of fake label numbers, keeping the analyst blind to the experimental groups.

### 2.8. Statistical Analysis

Shapiro–Wilk test was used to check the normality of units. One-Way ANOVA (analysis of variance) was carried out for micro-CT, pull-out, and histopathology results. Additionally, due to a *p*-value of <0.05 for pull-out results, the paired comparison of Tukey’s post hoc test was conducted between all three groups in the pull-out test.

## 3. Results and Discussion

Extracorporeal shock wave therapy can ease the understanding of osteointegration by regulating the immune response, encouraging regeneration of the jaw and alveolar bone, and endorsing angiogenesis and bactericidal effectiveness [19,20].

In our study, the mean body weight of animals was increased from 2.34 kg on surgery day to 2.56 on the sacrificing day. No difference in physical activity was observed between SWT-treated legs and the non-treated ones.

Micro-CT scan analysis was reported using bone volume fraction (BV/TV × 100). Total volume (TV) refers to the 3D volume of interest, which consists of screw, pores, screw-bone interface, and newly regenerated bone. While BV (Bone Volume) only refers to the newly regenerated bone in the 3D volume of interest (Figure 3).

Table 1 shows the descriptive statistics indicators and One-Way ANOVA (analysis of variance) for bone volume fraction based on micro-CT scan.

According to Table 1, the mean ± standard deviation of bone volume fraction in the control group was 31.75 ± 5.10, while in the one-time treated group and the three-time treatment group were equal to 33.83 ± 5.03 and 35.93 ± 3.21, respectively. This difference was not significant (*p*-value = 0.46).

Micro-CT method detects pre-existing differences between samples. Then, even samples of normal tissue may display microstructural inhomogeneities. The error can be occured by selecting only 2–3 histological sections for quantification. This can be solved by sampling more sections. However, increasing the number of sections meaningfully not be probable because processing sections includes cutting approx. 1 mm-thick plates of bone and losing part of the material during cutting and grinding. Micro-CT method is ran based on absorption, and it is mostly well adjusted for tissues such as bone but lacks sensitivity to image soft tissues. However, the ability of micro-CT to differentiate between fully mineralized and partially mineralized bone tissue is still restricted and histological sections should be applied for this purpose instead [26].

According to Table 2, the mean ± standard deviation of pull-out in the control group was 89.85 ± 7.57 and in the one-time treated and three-time treated groups were 96.36 ± 5.58 and 103.60 ± 0.53, respectively. This difference was statistically significant (*p*-value = 0.019).

Table 3 shows the results of paired comparison of Tukey’s post hoc test. Based on the results of Tukey’s test, the mean difference of pull-out between the control group and the three-time treated group was statistically significant (*p*-value = 0.015), while the mean difference between the other groups in pairs (the control group and the three-time treated group, the one-time treated group and the three-time treated group) was not statistically significant.

Histopathological findings are presented in Figure 4, which confirmed an increase in bone formation by ESWT. There was normal histology of bone matrix, Volkmann’s canals, osteocytes (lacunae containing osteocytes), and osteoblasts associated with no inflammation and bone necrosis in all experimental groups. However, there was a remarkable difference in the new bone formation degree between all groups (*p* < 0.05). Both shock wave-treated screws showed more regenerated bone compared to the control group. Indeed, the new bone formation was more severe in three-time treated screws compared to one-time treated screws. We did not find an immature bone in the control group, however, there were mature and immature bone formations simultaneous in three-time treated screws.

There are four phases for ESWT’s tissue affecting according to the litreture including physical, physicochemical, chemical, and biological phases. Shockwaves origin a positive pressure to produce absorption, reflection, refraction, and transmission of energy to tissues and cells in the physical phase that leads to the more permeability of cell membranes and ionization of biological molecules. Then, ESWT activates the release of biomolecules such as adenosine triphosphate (ATP) for the beginning of cell signaling paths in the physicochemical phase. In the third phase, the chemical phase, ESWT changs the functions of ion channels in cell membranes and mobilization of calcium. In the final phase, the biological phase, ESWT plays its part in moderating angiogenesis, anti-inflammatory properties, and healing of bone and soft tissue wounds [12].

Overall, the biomechanical pull-out test and histopathologic analysis both showed that ESWT enhanced bone regeneration at the screw-bone interface compared to the control group. Moreover, Tuckey’s post hoc findings for pull-out test affirmed that the three-time treated screws needed statistically significantly more force to be pulled out of the bone, compared to the other two groups.

In addition, the three-time treated group was the only group in which immature and mature bone formation zones were seen simultaneously; this suggests that if more time had been given to animals before sacrificing and after the last phase of ESWT, the outcome would have been even more perceptible in this group.

It is noteworthy that micro-CT analysis results did not show a statistically significant difference in favor of enhanced bone formation in shock wave-treated screws.

With this rising demand for dental implants in the last decades, it is a necessity to use new methods to obtain the best out of dental implants. This experiment along with previous animal studies, suggests that ESWT could be used as an adjuvant therapy to facilitate osseointegration of titanium implants in terms of time. This adjutancy would be even more substantial when it comes to fresh socket implants or immediate load implants, in which gaining the ultimate osseointegration in the shortest possible time is crucial [27,28]. Song et al. hypothesized that ESWT can regulate the activity of inflammatory cells, osteoblasts, and mesenchymal stem cells. They also concluded that ESWT can ease the recognition of osteointegration by regulating the immune response, inducing regeneration of the jaw and alveolar bone, and promoting angiogenesis and bactericidal efficacy [20].

Koolen et al. assessed the local shock wave treatment on osseointegration and subsequent screw fixation in rat bone with extracorporeal shock waves. Their results showed that four weeks after treatment, treated legs showed enhanced bone formation and screw fixation around cortical screws as compared to the control group. The authors verified this through an increased pull-out of the shock wave treated cortical screws. Development of de novo bone in the bone marrow was detected in some animals [24].

## 4. Conclusions and the Future Perspective

In conclusion, ESWT has characteristics of a truly non-invasive, safe and cost-beneficent adjuvant treatment to be added to dental implant procedures. Nevertheless, to this date, no human clinical trials have been conducted to evaluate the effect of ESWT in dental implants. Therefore, the clinical application of ESWT in dental implants requires further clinical trials.

There are a rare number of homogenous studies about ESWT, however, it has exposed many benefits and potential aids. There has not been a study of extraordinary difficulties, substantial injury, or tissue damage in the reported studies. It has been shown that ESWT established regenerative, anti-inflammatory, antibacterial, and analgesic properties. The type of ESW, dose, energy flux density, and a number of cycles need to be more optimized in the future. Investigators and clinicians should try to start and normalize the road map for homogenous predictable ESWT procedures in the oral and maxillofacial fields. Different parameters including physical and dose–effect are from the main establishing probable positive biological and regenerative regimens. Then, more clinical trials are wanted to control the perfect quantity of energy, dosage, and a number of visits essential for the best therapeutic consequences. More studies should be started on manufacturing suitable equipment and tools that match with the anatomical complexity of the oral and maxillofacial area. While numerous issues essential to be resolved in the future, extracorporeal shock wave therapy is assured to be an effective and suitable therapeutic technique to advance the dental implantation success rate and develop clinical implant suggestions.

It is also suggested that the large-animal models be considered for jaw bone implantation with weight-bearing stress. The observation period can be more prolonged to more influentially sense the effect of ESWT. The other studies in the future can be included the osseointegration of implants in unusual microenvironments, such as in diabetes or osteoporosis, and expounding the involvement result of ESWT on implants with poor initial stability or a poor healing state to afford a novel theoretic foundation for improving the success rate of dental implantation.

## 5. Study Limitations

There were several limitations to the current study. Some of the parameters (such as sample thickness) may have been affected by minor differences in sample orientation. Pictures from the micro-CT were evaluated by a different observer than the one who evaluated the pictures of histological sections. Therefore, the borders of regions of interest could differ to some extent. The difference in images can be occured also due to the different intensity of ossification at the bone surface in each sample, which causes the bone surface to be closer to the screw neck. Further, not having the same angle for the CT scan cuts can cause eye errors. The number of animals in the study were small. In the micro-CT technique the differences between samples may show microstructural inhomogeneities and selecting only 2–3 histological sections for quantification may lead to the error.

## Figures and Tables

**Figure 1 biomimetics-08-00137-f001:**
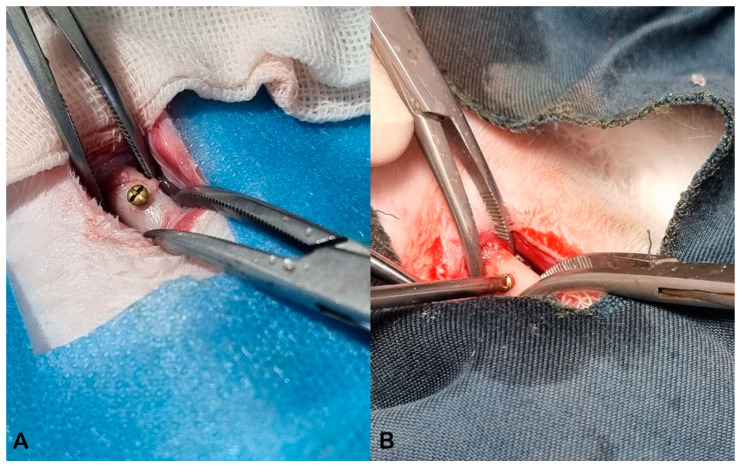
(**A**,**B**): insertion of the screw in rabbit femur. It was made sure that all screws were inserted unicortically with a proper primary stability.

**Figure 2 biomimetics-08-00137-f002:**
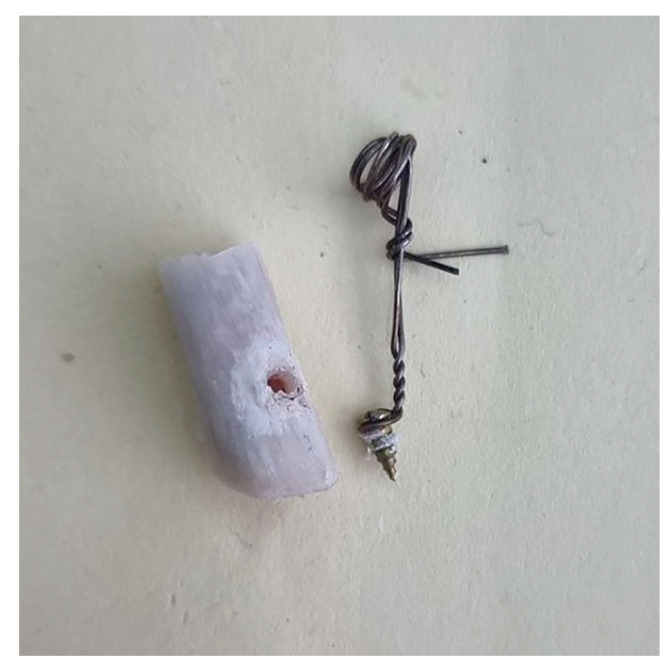
After pulling out the screw from the bone. Newly regenerated bone is visible on the screw threads.

**Figure 3 biomimetics-08-00137-f003:**
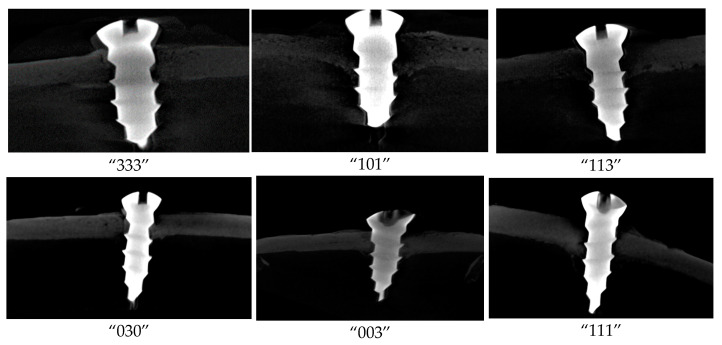
Micro computed tomography cuts of samples. The number of samples have been written under each sample. Radiolucencies around screw threads are due to drilling size, which gave us a slight micro defect around screws to assess the bone regenerating process.

**Figure 4 biomimetics-08-00137-f004:**
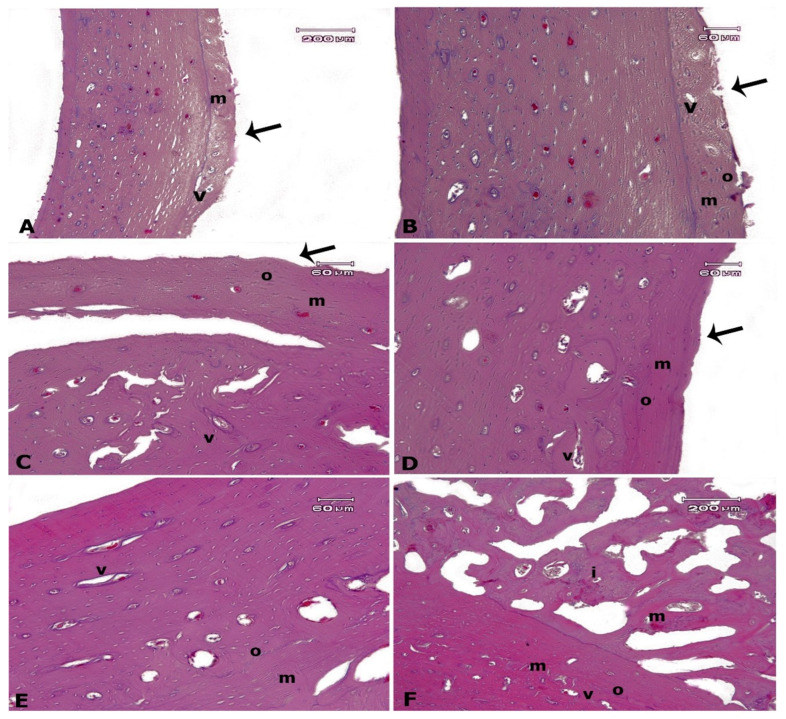
Rabbit histology of bone formation. (**A**,**B**): control group; (**C**,**D**): One-time treated group; (**E**,**F**): Three-time treated group. There was normal histology of bone matrix (m), Volkmann’s canals (v), and osteocytes (o), associated with no inflammation and bone necrosis in all experimental groups. Arrows are indicating bone formation zones, which were more severe in Three-time treated group compared to One-time treated group. There were mature and immature bone formations simultaneous in Three-time treated group (H&E). Regions of active ossification and newly formed bone are indicated by arrows in subfigure (**A**–**D**).

**Table 1 biomimetics-08-00137-t001:** Bone volume fraction in all three groups.

	N	Mean	Std. Deviation	*p*-Value
Control Group	4	31.75	5.10	0.46
One-time treated group	4	33.83	5.03	
Three-time treated group	4	35.93	3.21	

**Table 2 biomimetics-08-00137-t002:** Descriptive statistics indicators for Pull-out test in all three groups.

	N	Mean	Std. Deviation	*p*-Value
Control Group	4	89.85	7.57	0.019
One-time treated group	4	96.36	5.58	
Three-time treated group	4	103.60	0.53	

**Table 3 biomimetics-08-00137-t003:** Tukey’s post hoc test for pull-out results in all three groups.

Group		Mean Difference (I-J)	Std. Error	*p*-Value
(I)	(J)			
Control group	One-time treated group	−6.51	3.85	0.260
	Three-time treated group	−13.75	3.85	0.015
One-time treated group	Three-time treated group	−7.24	3.85	0.199

## Data Availability

The raw/processed data required to reproduce these findings can be shared after publication by requesting from the corresponding author.

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
