# Peer review of "Extracorporeal Shock Wave Therapy as a Helpful Method for Rapid Osseointegration of Dental Implants: Animal Study"

_biomimetics, 2023, doi:10.3390/biomimetics8020137_

Round 1

Reviewer 1 Report

Dear authors,

your article “Extracorporeal shock wave therapy as a helpful method for rapid osseointegration of dental implants: animal study” proposes an interesting method to increase osseointegration around implants.

However, it has some important methodology errors.

Materials and methods

Micro CT-scan

The tomographic images clearly show a difference in the insertion depth of the implants, and this factor leads to an incorrect standardization of the study and consequently inconsistent values in the Pull-Out test.

Pull-out test

The Pull-Out test procedure is incorrect, since the extraction force of the fixtures is not axial but oblique, and this factor influences the values obtained through the test.

Furthermore, more importantly, it is certainly wrong to perform the histologies in sites previously subjected to the Pull-Out test, as the extraction of the screws involves the destruction of the bone-implant interface and the impossibility of correctly interpreting the histologies. Other studies that performed this technique (Haas 1998, 2003) evaluated two surgical sites: one for the Pull-Out test and one for histology.

Histopathological analysis

The histological images clearly present the problem described above: it is impossible to record the bone-implant contact area and to define which area of the implant site the histologies refer to.

References

In the References the indications of the journals of publication are missing.

For these reasons, I believe that this article cannot be accepted for publication in Biomimetics in its present form.

Best regards

Reviewer 2 Report

Comments on the paper:

The manuscript titled: "Extracorporeal shock wave therapy as a helpful method for rapid osseointegration of dental implants: animal study" provides interesting findings and is worth of publication. I kindly suggest some additional corrections to make paper ready for publication.

1.    In the line 32, please replace "." after "results" and before "Additionally" with ","  and insert "," after "< 0.05" and before "for Pull-Out results", to get the single sentence.

2.    In the line 35, please correct the end of the sentence with .... "and the three-time treated group had the highest amount" .... of "what? "

3.    In the line 58, please insert "," after "rapidly" and "and shortening"

4.    In line 90, I suggest to replace "Dissimilarity to" with "Unlike"

5.    In line 110, the sentence "Three-time  treated group.... " is unclear regarding the day of the treatment and the day after the operation, please rewrite the sentence to make the procedure more clearer for readers.

6.    In line 217, please replace "3/21" with "3.21"

7.    In line 230, please rewrite the second part of the sentence "...., while the mean difference between the other groups was not statistically significant." , "the other groups" sounds confusing, although when looking in the Table 3 it becomes clear.

8.    In line 239, please correct the "both" to "Both.... "

9.    From the line 265 to 276, it seems to me to be more appropriate for that part of  "Results and Discussion" to be part of  "Introduction"

Author Response

The manuscript titled: "Extracorporeal shock wave therapy as a helpful method for rapid osseointegration of dental implants: animal study" provides interesting findings and is worthy of publication. I kindly suggest some additional corrections to make the paper ready for publication.

Thanks for your valuable comments. We modified the manuscript. The green highlights are related to the general modification and the blue highlights related to English language modifications.

  1. In line 32, please replace "." after "results" and before "Additionally" with "," and insert "," after "< 0.05" and before "for Pull-Out results", to get the single sentence.

RES: Thanks a lot. It has been done.

  1. In line 35, please correct the end of the sentence with .... "and the three-time treated group had the highest amount" .... of "what? "

RES: Thanks a lot. It has been corrected.

  1. In line 58, please insert "," after "rapidly" and "and shortening"

RES: Thanks a lot. It has been done.

  1. In line 90, I suggest replacing "Dissimilarity to" with "Unlike"

RES: Thanks a lot. It has been done.

  1. In line 110, the sentence "Three-time treated group.... " is unclear regarding the day of the treatment and the day after the operation, please rewrite the sentence to make the procedure clearer for readers.

RES: Thanks a lot. It has been corrected.

  1. In line 217, please replace "3/21" with "3.21"

RES: Thanks a lot. It has been done.

  1. In line 230, please rewrite the second part of the sentence "...., while the mean difference between the other groups was not statistically significant." , "the other groups" sounds confusing, although when looking in the Table 3 it becomes clear.

RES: Thanks a lot. It has been corrected.

  1. In line 239, please correct the "both" to "Both.... "

RES: Thanks a lot. It has been corrected.

  1. From lines 265 to 276, it seems to me to be more appropriate for that part of "Results and Discussion" to be part of the "Introduction"

RES: Thanks a lot. It has been moved to the introduction section.

Reviewer 3 Report

The work is pertinent and the design of this one seems to me to be well elaborated. The use of this technique appears to be promising in increasing the integration of implants. However, for this work to be accepted for publication, several changes must be made. Although the experimental work is preliminary and considering the evaluation of 2 variables, the number of animals in the study seems to me to be too short to evaluate the results with statistical certainty. The description of the mechanical tests should be better explained and better demonstrated by images. The same goes for the histomorphometric evaluation, which will have to be clearer and more descriptive with more images of the test groups, so that whoever sees the images can clearly see what is described in a very confusing way. The structure of the work looks good to me, but the discussion is clearly poor and should be substantially improved and supported with more bibliography to be inserted in this chapter. Finally, English will also need to be improved. There are also some spelling errors.

Author Response

The work is pertinent and the design of this one seems to me to be well elaborated. The use of this technique appears to be promising in increasing the integration of implants. However, for this work to be accepted for publication, several changes must be made.

Although the experimental work is preliminary and considering the evaluation of 2 variables, the number of animals in the study seems to me to be too short to evaluate the results with statistical certainty.

Thanks for your valuable comments. We modified the manuscript. The green highlights are related to the general modification and the blue highlights related to English language modifications.

RES: Thanks for your valuable comment. Due to the limitations of our animal providing organization, we could not use more animals exclusively for histopathology analysis. We used ShapiroWilk test for determining normality and the data were normal.

However, we added this point in our limitations.

The description of the mechanical tests should be better explained and better demonstrated by images. The same goes for the histomorphometric evaluation, which will have to be clearer and more descriptive with more images of the test groups, so that whoever sees the images can clearly see what is described in a very confusing way.

RES: Thanks a lot. We improved the results section and the figures.

The structure of the work looks good to me, but the discussion is clearly poor and should be substantially improved and supported with more bibliography to be inserted in this chapter.

RES: Thanks a lot. We improved the discussion.

Finally, English will also need to be improved. There are also some spelling errors.

RES: Thanks a lot. We improved the language. The blue highlights are related to the English corrections.

Reviewer 4 Report

The purpose of this study was to evaluate the effect of focused extracorporeal shock wave therapy on osseointegration of titanium mini-screws as a representative of dental implants on cortical rabbit bone was tested. In addition, the effect of treatment series and treatment intervals were also studied. This study holds merit and the authors deserve recognition for their efforts in conducting this study. However, the authors should address some concerns before we can accept the article for publication.

1.     The abstract lacks supporting evidence. The authors must provide sufficient quantitative data to support their claims.

2.     Line 188, “One-Way ANOVA (analysis of variance) was done for Micro-CT, Pull-Out, and histopathology results.”. However, the sample size of this study is small, so the nonparametric analysis may be more suitable.

3.     The authors must expand the “Discussion” section by thoroughly interpreting all their experimental results and including an exhaustive comparison between their experimental results and the existing literature.

4.     In the section "Biomechanical pull-put test", please provide more detail about this experiment, e.g. loading speed (strain rate), and load cell size.

5.     In the section "miccro-CT", please provide more detail about this experiment, e.g. scanning voltage, current, and resolution.

6.     Please consider adding more description about the relation between primary stability and osseointegration.

Author Response

The purpose of this study was to evaluate the effect of focused extracorporeal shock wave therapy on osseointegration of titanium mini-screws as a representative of dental implants on cortical rabbit bone was tested. In addition, the effect of treatment series and treatment intervals were also studied. This study holds merit and the authors deserve recognition for their efforts in conducting this study. However, the authors should address some concerns before we can accept the article for publication.

The abstract lacks supporting evidence. The authors must provide sufficient quantitative data to support their claims.

Thanks for your valuable comments. We modified the manuscript. The green highlights are related to the general modification and the blue highlights related to English language modifications.

RES: Thanks a lot. We improved the abstract.

Line 188, “One-Way ANOVA (analysis of variance) was done for Micro-CT, Pull-Out, and histopathology results.”. However, the sample size of this study is small, so the nonparametric analysis may be more suitable.

RES: Thanks a lot. We used ShapiroWilk test for determining normality and the data were normal.

The authors must expand the “Discussion” section by thoroughly interpreting all their experimental results and including an exhaustive comparison between their experimental results and the existing literature.

RES: Thanks a lot. We improved the discussion.

In the section "Biomechanical pull-put test", please provide more detail about this experiment, e.g. loading speed (strain rate), and load cell size.

RES: Thanks a lot. It has been done.

In the section "miccro-CT", please provide more detail about this experiment, e.g. scanning voltage, current, and resolution.

RES: Thanks a lot. It has been done.

Please consider adding more description about the relation between primary stability and osseointegration.

RES: Thanks a lot. It has been done. We added some data into the introduction section in this regards.

Reviewer 5 Report

Dear authors !

I was given the opportunity to review Your manuscript presenting the results of Your experimental in vivo study investigating the outcome of the application of ESWT in the course of osseointegration of titanium implants in an animal model.

The experimental setup and execution is concise, clear, reproducible and flawless. The procedural choice of oversized drilling before implant insertion is excellent and increases the significance and validity of the results greatly.

References are up-to-date and statistical methods used for evaulation appropriate.

Although the manuscript is written very comprehensively also for readers not specialized in the biology of bone-regeneration, this reviewer found major issues in English syntax and especially wording.

The authors are kindly advised (although this reviewer perfectly knows about the current critical general situation of academic exchange) to revise the manuscript with the aid of a native English speaker regarding English syntax and wording. It would be a pitty if this excellent research-paper could not be published due to language-problems.

Thank You 

Author Response

I was given the opportunity to review Your manuscript presenting the results of Your experimental in vivo study investigating the outcome of the application of ESWT in the course of osseointegration of titanium implants in an animal model.

The experimental setup and execution is concise, clear, reproducible and flawless. The procedural choice of oversized drilling before implant insertion is excellent and increases the significance and validity of the results greatly.References are up-to-date and statistical methods used for evaulation appropriate.Although the manuscript is written very comprehensively also for readers not specialized in the biology of bone-regeneration, this reviewer found major issues in English syntax and especially wording.The authors are kindly advised (although this reviewer perfectly knows about the current critical general situation of academic exchange) to revise the manuscript with the aid of a native English speaker regarding English syntax and wording. It would be a pitty if this excellent research-paper could not be published due to language-problems.

RES: Thanks a lot. We improved the language. The blue highlights are related to the English corrections.

 We modified the manuscript. The green highlights are related to the general modification and the blue highlights related to English language modifications.

Round 2

Reviewer 1 Report

Dear authors,

your article has certainly been improved with the changes and corrections you have made.

The procedural errors that I pointed out to you in the previous review remain and which I hope you will take into account in a future work.

However, considering the originality of the article, I think it deserves to be published in "Biomimetics" in this form.

Best regards

Reviewer 3 Report

I think the paper is now in conditions to be accept